# Inconsistent treatments of the kinetics of Clustered Regularly Interspaced Short Palindromic Repeats (CRISPR) impair assessment of its diagnostic potential

## Perspective

**Key words:**
Clustered Regularly Interspaced Short Palindromic Repeats; CRISPR; CRISPR-Cas; enzyme kinetics; limit of detection; molecular diagnostics

**Author for correspondence:**
*Juan G. Santiago,
E-mail: juan.santiago@stanford.edu

## Juan G. Santiago* [ID]

Department of Mechanical Engineering, Stanford University, Stanford, CA 94305, USA

## Abstract

The scientific and technological advent of Clustered Regularly Interspaced Short Palindromic Repeats (CRISPR) is one of the most exciting developments of the past decade, particularly in the field of gene editing. The technology has two essential components, (1) a guide RNA to match a targeted gene and (2) a CRISPR-associated protein (e.g. Cas 9, Cas12 or Cas13) that acts as an endonuclease to specifically cut DNA. This specificity and reconfigurable nature of CRISPR has also spurred intense academic and commercial interest in the development of CRISPR-based molecular diagnostics. CRISPR Cas12 and Cas13 orthologs are most commonly applied to diagnostics, and these cleave and become activated by DNA and RNA targets, respectively. Despite the intense research interest, the limits of detection (LoDs) and applications of CRISP-based diagnostics remain an open question. A major reason for this is that reports of kinetic rates have been widely inconsistent, and the vast majority of these reports contain gross errors including violations of basic conservation and kinetic rate laws. It is the intent of this *Perspective* to bring attention to these issues and to identify potential improvements in the manner in which CRISPR kinetic rates and assay LoDs are reported and compared. The CRISPR field would benefit from verifications of self-consistency of data, providing sufficient data for reproduction of experiments, and, in the case of reports of novel assay LoDs, concurrent reporting of the associated kinetic rate constants. The early development of CRISPR-based diagnostics calls for self-reflection and urges us to proceed with caution.

## Introduction

One of the most exciting developments of the past decade is gene editing made possible by the technology called CRISPR, an acronym for Clustered Regularly Interspaced Short Palindromic Repeats. This advance has recently been celebrated by the award of the 2020 Nobel Prize in Chemistry to Emmanuelle Charpentier and Jennifer Doudna for their work on CRISPR-Cas9. This technology involves two essential components, (1) a guide RNA to match a targeted gene and (2) Cas 9, Cas12 or Cas13, which is a CRISPR-associated protein that acts as an endonuclease that cuts DNA. All these activities of the CRISPR-Cas system are based on its target-specific binding, allowing it to be applied for the development of diagnostic methods for detecting disease-related genes as well as microRNAs and the genetic variations such as single nucleotide polymorphism (SNP) and DNA methylation. For example, CRISPR diagnostics are viewed as a serious contender for fast and deployable assays for the detection of the RNA of SARS-CoV-2, the virus that causes Covid-19 (Nguyen *et al.,* 2020; Ramachandran *et al.,* 2020; Fozouni *et al.,* 2021; Shinoda *et al.,* 2021). The most commonly applied Cas types for diagnostics are Cas12 and Cas13, which offer direct detection of DNA and RNA targets, respectively (Tang *et al.,* 2021).

As a diagnostic tool, CRISPR-Cas systems offer distinct advantages as they can be reconfigured using synthetic guide RNAs, are compatible with a few detection modalities including fluorescent reporter probes, are compatible with lyophilised reagents and will likely be field deployable, particularly given robust enzyme activity at ambient temperatures (Tang *et al.,* 2021). This potential has also led to significant investments in new biotechnology companies pursuing CRISPR diagnostic as products, notably Mammoth Biosciences, Inc. in Brisbane, CA and Sherlock Biosciences in Cambridge, MA. The ultimate limits of detection (LoDs) of the CRISPR assay itself (e.g. without amplification), however, are fundamentally limited by the kinetic rates the enzymes can achieve (Ramachandran and Santiago, 2021). In this regard, there appears to be much confusion in the literature with discordant estimates provided, as will be discussed in what follows. The purpose of this *Perspective* is point out potential improvements in how CRISPER kinetic rates are calculated and how the LoDs of assays are reported and compared.

## Sensitivity of CRISPR/Cas systems

CRISPR enzymes are understood to be highly specific (Chen *et al.,* 2018; Ooi *et al.,* 2021), but the ultimate LoDs of associated assays are and will continue to be fundamentally limited by the kinetic rates the enzymes can achieve (Ramachandran and Santiago, 2021). Current CRISPR applications envision a two-step assay composed of an initial *cis*-cleavage process where a programmed (via guide RNA molecule) CRISPR enzyme specifically detects a target sequence, cleaves it and becomes activated as a result of this interaction (Gootenberg *et al.,* 2017; Chertow, 2018; Kellner *et al.,* 2019). The timescale for completion of this first step is relatively short (Jeon *et al.,* 2018) and approximately inversely proportional to the initial enzyme concentration. The second assay step, the *trans*-cleavage depicted in Fig. 1, is therefore the rate-limiting step. In this second step, the now-activated enzyme indiscriminately cleaves reporter molecules, most commonly synthetic ssDNA or ssRNA nucleic acids functionalised with a fluorophore and quencher on each end.

For the present, the classic Michaelis–Menten kinetics model seems to offer the best framework to quantify CRISPR-Cas enzymatic rates. This quantification can be summarised with two key parameters, the enzyme turnover rate, $k_{cat}$, and the catalytic efficiency, $k_{cat}/K_m$, where $K_m$ is the Michaelis–Menten constant. $K_m$ describes an equilibrium of activated enzyme and reporter in terms of a ratio of the sum of the reporter dissociation rate $k_r$ and of $k_{cat}$ and the forward association associate rate $k_f$, as indicated in the equation in the figure (Chen *et al.,* 2018; Slaymaker *et al.,* 2019; Ramachandran and Santiago, 2021). There has been, appropriately, enormous interest in the quantification of CRISPR kinetics rates (and hence achievable sensitivity and speed) of these systems, particular for the Cas12 and Cas13 orthologs (Chen *et al.,* 2018; Li *et al.,* 2019; Slaymaker *et al.,* 2019).

## Inconsistent treatments of CRISPR kinetics

Chen *et al.* (2018) and Slaymaker *et al.* (2019) are the two seminal papers on CRISPR kinetics. Indeed, these papers are the first to present quantitative kinetic rate data of any kind and so mark the start of truly quantitative studies of CRISPR diagnostics. The principal investigators and communicating authors of these two publications are two world leaders in the field, the aforementioned Jennifer Doudna of UC Berkeley (and co-founder of Mammoth Biosciences) and Feng Zhang of MIT (and co-founder of Sherlock Biosciences), respectively. These first two papers set a high bar for enzyme kinetic performance and identified Cas12a and Cas13b as diffusion-limited enzymes with turnover rates as high as 1,250 and 1,000 s$^{-1}$, respectively. Since that time, the international field has many times apparently 'corroborated' these impressive CRISPR

rate measurements, and even striven to find and report modifications that reportedly yield ever higher kinetic rates.

In early 2020, my then PhD student Ashwin Ramachandran and I began work on applying CRISPR enzyme kinetics to microfluidic assays. We were encouraged by CRISPR's exceedingly high and 'corroborated' kinetic rates. In March 2020, we focused on detection of the RNA of SARS-CoV-2. (My lab also pursued, and continues to pursue, increased CRISPR reaction rates by preconcentration of reagents using electric fields (Ramachandran *et al.,* 2020).) We approached the problem as engineers, and so began by developing analytical and numerical models for CRISPR kinetics assuming the Michaelis–Menten framework. This included exercises such as inputting published values of $k_{cat}$ and $k_{cat}/K_m$ into our models and then simulating data such as progress curves. We also initiated experimental studies of CRISPR kinetics, and this eventually led to our posing (Ramachandran and Santiago, 2021) of the following three back-of-the-envelope checks of self-consistency of enzyme kinetic rate experimental data:

$$\alpha = \frac{v t_{\text{lin}}}{S_0}; \tag{1}$$

$$\beta = \frac{v}{k_{cat} E_0}; \tag{2}$$

$$\gamma = \frac{t_{\text{lin}} k_{cat} E_0}{K_M}. \tag{3}$$

Here, $S_0$ and $E_0$ are, respectively, the reported initial concentrations of reporters (i.e. enzyme substrate) and of activated enzyme. Due to the nature of *cis*-cleavage, the maximum achievable value of $E_0$ is believed to be equal to the target concentration. $v$ is the reported velocity of reaction (i.e. rate of production of cleaved reporters) and can usually be obtained from the initial slope of progress curves. $t_{\text{lin}}$ is the timescale of the initial linear phase of signal increase. The latter parameter is especially useful as it can be estimated from raw data without knowledge of calibrations and even with incorrect calibration, so long as some measure of signal is reported as a function of (the correct) timescale. Briefly, $\alpha$ must be strictly less than unity in accordance with conservation of species. That is, the number of reporters cleaved in the initial linear portion of the progress curve simply must be less than total number of reporters in the (closed) system. $\beta$ should be strictly less than unity in accordance with the fact that the reported velocity must be less than the theoretical maximum value within the Michaelis–Menten framework, $k_{cat} E_0$. Accordingly, $v$ can be obtained from the slope of one or more progress curves for which the initial substrate concentration, $S_0$, was significantly higher than the reported value of $K_M$. $k_{cat} E_0$ is then independently determined from the reported value of

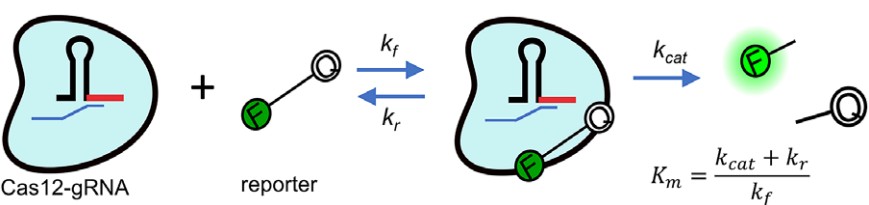

**Figure 1.** Michaelis–Menten kinetics model of *trans*-cleavage of activated CRISPR-Cas12. The Cas12 (blue blob) is functionalised with a synthetic guide RNA molecule (shown here as a hairpin structure). In this rate-limiting *trans*-cleavage step, the enzyme has been functionalised by recognition and cleavage of the target DNA molecule (cleaved ssDNA target shown as a small blue strand). The enzyme indiscriminately cleaves ssDNA. By design, stoichiometry then favours cleaving of nucleic acid reporter probes functionalised with a fluorophore (*F*) and quencher (*Q*). $K_m$ is the Michaelis–Menten constant defined in terms of the off-rate $k_r$, turnover rate $k_{cat}$ and forward rate $k_f$, as shown in this figure.

$k_{cat}$ and the reported target concentration (as the maximum possible value of $E_0$).

For self-consistency, $\gamma$ should also be less than order unity. The idea behind this third ratio is that the timescale of the linear portion of the reaction should be less than an estimate of the timescale for completion of the reaction. In Ramachandran and Santiago (2021), we considered for the latter estimate the case of a reaction, wherein $S_o$ is significantly lower than the value of $K_M$. For this case, there is a simple first-order decaying exponential solution for the substrate concentration with a timescale of $K_M/(k_{cat}E_0)$. The latter regime describes a condition likely to be employed in a practical bioassay based on CRISPR-Cas since reporter concentration should be kept relatively low to avoid the significant background signal associated with uncleaved reporters (Huyke et al., 2022). However, the formulation for $\gamma$ is nevertheless fairly generally valid. For example, consider the regime where $S_0$ is significantly *higher* than the value of $K_M$. In this regime, the reaction velocity can be simplified as follows:

$$\frac{d[P]}{dt} = k_{cat}E_0 \frac{[S]}{K_M + [S]} \approx k_{cat}E_0. \tag{4}$$

Here, $[P]$ is the concentration of cleaved reporters and the first equal sign is consistent with the Michaelis–Menten quasi-steady state and reactant stationary assumptions (Schnell, 2014). The second approximate equality is for the abundant substrate regime. For such a regime, $[P]$ grows linearly while reporter concentration $[S]$ decreases linearly as $[S] \approx S_0 - k_{cat}E_0t$. Hence, the timescale for reporter concentration to drop from its initially high value to a value of order $K_M$ is simply approximately $(S_0 - K_M)/(k_{cat}E_0)$. Thereafter, the reaction rate becomes self-limiting as reporters are consumed; hence, the subsequent stage of the reaction should scale with an additional time of order $K_M/(k_{cat}E_0)$. We can combine these arguments into a simple heuristic which provides a new formulation of $\gamma$ valid for abundant reporters, wherein the total reaction timescale is of order $K_M/(k_{cat}E_0) + (S_0 - K_m)/(k_{cat}E_0)$ as follows:

$$\gamma' \approx \frac{t_{lin}k_{cat}E_0}{S_0}, \tag{5}$$

where the prime indicates a heuristic formulation for the $\gamma$-type parameter for the case of abundant reporters. $\gamma'$ (Eq. (5)) should be less than unity for kinetic rate data for which $S_0 \gg K_m$. Note that, in the latter regime, $\gamma$ is necessarily larger than $\gamma'$, hence all Michaelis–Menten kinetic rate data (irrespective of the relative value of $S_0$ and $K_m$) should have $\gamma$ (Eq. (3)) less than about unity.

In the fall of 2020, we applied the criteria that the expressions of Eqs. (1)–(3) should be less than unity to all existing CRISPR kinetics papers and became convinced of an astounding fact: All but one (Cofsky et al., 2020) of the eight reported sets of data (Chen et al., 2018; Shan et al., 2019; Slaymaker et al., 2019; Cofsky et al., 2020; Nguyen et al., 2020; Zhou et al., 2020) as of that date across the entire field of CRISPR diagnostics presented kinetics data that upon close inspection exhibited gross inconsistencies, including gross violations of basic laws of mass conservation and chemical kinetics (Ramachandran and Santiago, 2021). The single paper (Cofsky et al., 2020) that presented self-consistent data was an outlier, reporting extremely low kinetic rates. Further, the latter paper studied variations of the Cas12a system and carefully cited one of the aforementioned papers (Chen et al., 2018) as describing the kinetic rates achievable with

Cas12a (and shared the same communicating author, J. Doudna, with the latter 2018 paper).

We communicated our results to these labs and, most often, received no response. We eventually established very cordial personal communications with the authors of the two seminal papers in the field: Chen et al. (2018) and Slaymaker et al. (2019). To their great credit, these authors were very open minded and agreed to review the rate data published in their well-cited papers. Consequently, our pointing out of these gross errors eventually led to two published errata (Chen et al., 2021; Slaymaker et al., 2021) of these seminal works. Both teams cited errors associated with unit conversions as the cause of the gross discrepancies. We were very grateful to these careful researchers, particularly the two respective communicating authors Jennifer Doudna of Berkeley and Feng Zhang of MIT, who led the publication of the errata and kindly acknowledged our role in pointing out the discrepancies.

To our knowledge, no other published work in the field has published corrections of what seem to be obvious and gross discrepancies in CRISPR kinetics measurements (cf. Table 1 of Ramachandran and Santiago, 2021). Meanwhile, publication of grossly inconsistent data continues. For example, Shan et al. (2019) published what is, to my knowledge, the highest turnover rate for CRISPR to date: $k_{cat}$ = 4,854 for LbuCas13a (with an ssRNA activator). More recently, Yue et al. (2021) reported the second-highest value of $k_{cat}$ = 4,850, but for LbCas12a (with an ssDNA activator). These top, record-setting turnover rates match each other to three significant figures (0.08%). This is surprising as CRISPR enzyme kinetic rates such as $k_{cat}$ and $K_m$ vary significantly among orthologs, guide RNA, targets and buffer chemistry. Moreover, these quantities can be measured only to within a factor of 2 or so. More to the point, the publication by Yue et al. (2021) reports signal strength equivalent to about 90 times the maximum achievable value given the reported calibration and reporter concentration and signal data with a linear phase timescale that is about 380 times larger than the theoretical completion time for their reaction. The latter quantities correspond to values of $\alpha$ = 90 and $\gamma$ = 380, respectively (Ramachandran and Santiago, 2021).

The reasons for past and continued gross discrepancies in the CRISPR field are unknown. I believe that one cannot underestimate the immense influence of the two seminal papers of Chen et al. (2018) and Slaymaker et al. (2019) and the widespread influence of their respective principal investigators. Each of these publications concluded that CRISPR enzymes exhibited diffusion-limited rates. Doudna and Zhang, leaders in the field, published roughly similar values for turnover rates of order 1,000 s$^{-1}$ and the latter seemed to corroborate the former. These rates are now known to be incorrect by at least two orders of magnitude, but the initial reports likely contributed to selection bias in favour of high rates.

## Room for more consistent reporting and corroboration of data in the CRISPR kinetics field

The early development of CRISPR diagnostics suggests that the field would benefit from improved reporting and corroboration. Of course, corroboration requires that enough data and details be reported so that others can repeat experiments. I here offer the following suggestions:

1. Researchers should check for self-consistency in data. We have suggested one method which involves three simple, back-of-the-envelope calculations to validate the self-consistency of

CRISPR kinetics data (Ramachandran and Santiago, 2021), and we advocate the use of such sanity checks for all enzyme kinetics studies.

2. Publications should report data sufficient to reproduce the experiments. Interestingly, the vast majority of all publications around CRISPR kinetics and CRISPR diagnostic LoDs do not publish sufficient data to reproduce the work and enable fair comparisons. By far, the most commonly missing piece of information is the calibration data used to relate measured fluorescence signals (in arbitrary units) to cleaved and uncleaved reporter concentrations (note uncleaved reporters exhibit significant fluorescence signal) (Ramachandran and Santiago, 2021; Huyke et al., 2022). A second critical and commonly missing piece of information is quantification of the non-specific background activity of CRISPR enzymes (i.e. in the absence of target). This important drawback of these enzymes very likely limits their LoD (Huyke et al., 2022) and has been documented by a variety of groups (Palmier and Van Doren, 2007; Shan et al., 2019; Nguyen et al., 2020; Fozouni et al., 2021; Nalefski et al., 2021; Shinoda et al., 2021; Yue et al., 2021; Huyke et al., 2022).

3. Researchers should share raw data with others if asked. My group has found it difficult to secure responses or, in cases when there is response, to convince authors of CRISPR kinetics studies to share raw data. This holds even for papers published in journals whose publication agreements specifically require it. Sharing of raw data is an ethical duty of scientists.

4. Importantly, reports of CRISPR assays reporting LoDs (particularly amplification-free assays) should present at least rough measures of the enzyme kinetic rates required for such signals. This can be done by, for example, performing a set of simple parallel experiments to quantify $k_{cat}$ and $K_m$ using traditional equipment such as thermal cyclers.

5. Of course, authors of papers whose data exhibit serious discrepancies should follow the admirable example of Doudna and Zhang and publish corrections to their own data.

## Emerging CRISPR assays and LoDs

The CRISPR diagnostic field continues to grow and find application to a variety of diseases. Those that employ pre-amplification clearly offer the highest sensitivity and the subsequent CRISPR step can in some cases be used to improve specificity and provide a convenient fluorescent signal modality. The combination of CRISPR with isothermal pre-amplification such as loop-mediated isothermal amplification (Agrawal et al., 2020; Nguyen et al., 2020; Ramachandran et al., 2020; Ooi et al., 2021) and recombinase polymerase amplification (Gootenberg et al., 2018; Arizti-Sanz et al., 2020) particularly has potential towards point-of-care assays.

The number of and variety among CRISPR assays without amplification is growing rapidly with microfluidic chip formats (Bruch et al., 2019; Fozouni et al., 2021) including droplet or microchamber systems (Ackerman et al., 2020; Shinoda et al., 2021), lateral flow readout formats (Gootenberg et al., 2018) and detection using multiple types of Cas enzymes per target (Liu et al., 2021). However, the ultimate achievable sensitivity of CRISPR assays without amplification remains an open question. Reports of amplification-free assays uniformly do not provide sufficient information to check for self-consistency. We can, however, make estimates of the LoD of CRISPR-based enzyme

kinetics parameters (Huyke et al., 2022). For example, in the case of CRISPR assays with cleaved reporter fluorescence readout, the rate of signal production is, under the Michaelis–Menten framework, limited by either $k_{cat}$ or $k_{cat}/K_m$, depending on the relative values of $K_m$ and the reporter concentration used in the assay. In most assays, the need to limit background signal typically drives assay designs towards reporter concentrations on the order of or less than the value of $K_m$ and so $k_{cat}/K_m$ is likely the most important parameter (Ramachandran and Santiago, 2021). Once we estimate possible signal strength, we should compare this to the background signal level and to the reproducibility of that background level. For example, the synthetic, fluorophore-quencher ssRNA and ssDNA probes exhibit cleaved-to-uncleaved fluorescence signal ratios of only about 10 or less. Also, CRISPR enzyme systems typically have significant non-specific activity which significantly contributes to background signal. We recently evaluated the kinetic rates, background activity and observable LoDs of LbCas12a, AsCas12a, AapCas12b, Lwa-Cas13a and LbuCas13a (Huyke et al., 2022). We found detection of fluorescent reporters requires cleaving of at least about 0.1% of reporters in the reaction. For the kinetic rates we have observed in our lab, this translates to LoDs of order picomolar. This LoD is consistent with most of the CRISPR studies summarised by Kaminski et al. (2021).

Given the picture that is forming around CRISPR kinetic rates and the aforementioned estimates, it is difficult to reconcile many of LoDs currently being reported. Consider, for example, the order 50 aM sensitivity reported by Silva et al. (2021) for amplification-free detection of the reverse-transcribed cDNA of SARS-CoV-2 RNA using LbaCas12a. This sensitivity approaches that of amplified assays for this sample type (Agrawal et al., 2020; Arizti-Sanz et al., 2020) and was achieved using a camera from a cell phone (detecting a gas bubble associated with triggered catalase activity). The CRISPR kinetic rates responsible for such a strong signal-to-background ratio cannot be estimated because the required data (e.g. raw fluorescence progress vs time curves or Michaelis–Menten curves) are not included in that publication. As a second example, consider the work of Fozouni et al. (2021) who reported amplification-free detection of SARS-CoV-2 RNA with a LoD of order 100 aM from nasal swabs. The latter assay used two Lbu-Cas13a systems, a 400 nM reporter concentration and a mobile phone camera for detection in 30 min. Given our own quantification of LbuCas13a kinetics in similar buffers ($k_{cat}/K_m$ of order 4E6 $M^{-1}$ $s^{-1}$, Huyke et al., 2022) we estimate that the latter work reports detection after cleaving less than about 0.0001% of the fluorophore-quencher reporters in the reaction. Consider, for example, that the intact reporters in this assay would produce a signal equivalent to more than about 10% of cleaved reporters and that such a background is very difficult to reproduce within, say, 4% from experiment to experiment. The latter work presents progress curves showing signal growth versus time – but not sufficient data and calibration for a reader to estimate both $k_{cat}$ and $k_{cat}/K_m$. We do know that, as a corroboration of their observed strong signals and very high kinetic rates (including one reported turnover rate of 600 $s^{-1}$), the latter paper cites the kinetic measurements of Slaymaker et al. (2019). Unfortunately, as mentioned earlier, Slaymaker and colleagues have now corrected their turnover rate from 1,000 to 1 $s^{-1}$, and so this 'corroboration' for fast rates now seems less than useful. Even assuming the vastly overestimated, incorrect kinetic rates originally reported by Slaymaker, we estimate the Fozouni assay would cleave only 0.01% of reporters.

## Various advantages and disadvantages of CRISPR-Cas diagnostic assays

This section presents a brief and qualitative summary of the likely advantages and disadvantages of current CRISPR-Cas systems as candidate assays for molecular diagnostics (see also Tang *et al.,* 2021). CRISPR-Cas advantages clearly include the room temperature activity of the enzyme and compatibility with lyophilised reagents (Curti *et al.,* 2021). Importantly, the assays are highly reconfigurable to varying target sequences by simple variation of a portion of the synthetic guide RNA sequence. This reconfigurability also lends itself to various multiplexing approaches. For example, one can simply run many parallel reactions with varying guide RNA but the same, well-characterised CRISPR-Cas enzyme. Another advantage of CRISPR-Cas systems is the seemingly permanent state of an activated enzyme which, of course, offers amplification of signal, albeit most commonly a simple linear amplification of signal that is fundamentally limited by kinetic rates (Huyke *et al.,* 2022). The implementation of CRISPR-Cas assays with reporter molecules composed of synthetic nucleic acids functionalised with fluorophore and quencher pairs also offers a convenient detection modality in the form of an increase of a fluorescence signal. Other choices for detection include cleaving of biotinylated nucleic acid reporters followed by a separate lateral flow assay readout (Gootenberg *et al.,* 2018). Lastly, the most important advantage of current CRISPR-Cas is, in the opinion of this author, its high degree of specificity, including specificity to SNPs in the target molecule (Blanluet *et al.,* 2022).

CRISPR-Cas systems, in their current form, also exhibit some fairly strong disadvantages. Likely, the most important limitation is that the relatively low kinetic rates (Ramachandran and Santiago, 2021; Huyke *et al.,* 2022) of current CRISPR-Cas systems as these severely limit the sensitivity of assays which rely solely on the linear amplification of CRISPR-Cas enzymes to produce signal. This limitation implies that most, if not all, simple bulk reaction assays using CRISPR-Cas should rely on some sort of up-front (or simultaneous) exponential pre-amplification of target nucleic acid. Such exponential amplification may come from polymerase chain reaction or one of various available isothermal amplification methods. Improved sensitivity using amplification-free CRISPR-Cas assays may be possible by, for example, leveraging droplet- or microchamber-based, single-molecule sensing schemes (Shinoda *et al.,* 2021). Huyke *et al.* (2022) hypothesised that the latter schemes may benefit from a beneficial selection bias in favour of the highest activity molecules of an ensemble of single-molecule reactions and from the increase of signal intensity associated with the confinement of cleaved reporter molecules (to the single droplet in which they are cleaved). Electric-field-based focusing of CRISPR-Cas reagents into a small volume may also increase sensitivity by two or three orders of magnitude (Ramachandran *et al.,* 2020). Another exciting possibility for improved sensitivity may be devising reaction schemes based on CRISPR-Cas enzyme reactions which may exhibit positive feedback and hence exponential amplification of signal (e.g. see Shi *et al.,* 2021).

A second important limitation is that CRISPR-Cas enzymes have finite non-specific background activity such that they cleave nucleic acids (e.g. reporters) in the absence of the target sequence (Palmier and Van Doren, 2007; Shan *et al.,* 2019; Nguyen *et al.,* 2020; Fozouni *et al.,* 2021; Nalefski *et al.,* 2021; Shinoda *et al.,* 2021; Yue *et al.,* 2021; Huyke *et al.,* 2022). Such background activity establishes an important, and difficult-to-avoid, constraint on

LoD for each assay (Huyke *et al.,* 2022). CRISPR-Cas assays which use fluorophore-quencher pair reporters (the most common detection modality) further suffer from the fact that uncleaved, 'quenched' reporters have significant fluorescence signal. In our experience, cleaved-to-uncleaved fluorescence signal ratios of commercially available reporters are order 10 and this severely limits the dynamic range and LoDs of simple assays based on end-point signal detection. Also, since the precise starting value of reporter background signal can be difficult to reproduce across experiments and assays (e.g. due to varying degrees of degradation of reporters and/or even slight contamination with fluorescent species), this limits the sensitivity of end-point signal detection methods. CRISPR-Cas assays are also relatively novel and their sensitivity and specificity have not been sufficiently explored and optimised. Consider, for example, that kinetic rates depend strongly on the CRISPR-Cas ortholog; single versus double stranded nature of DNA target (e.g. for Cas12 orthologs); type, length and sequence of reporter molecules; gRNA length and sequence; enzyme preparation prior to assay (including incubation with gRNA); buffer chemistry including additives; temperature and other assay conditions. Lastly, there is a dearth of studies which quantify CRISPR-Cas kinetic rates (and sensitivity) and specificity in the presence of varying types and abundance of background nucleic acids and/or challenging sample matrix chemistries.

## Outlook

CRISPR enzyme systems are certainly one of the most important developments in biology in the last decade–particularly in the realm of gene editing. Reports of CRISPR-Cas molecular diagnostic assays suggest strong potential and impressive versatility across sample types and assay formats. However, as has been summarised in this *Perspective*, there remain significant challenges and important open questions. A major goal of this *Perspective* is to emphasize the importance of accurate and clear reporting of CRISPR-Cas kinetic rates and the intimately related issue of possible LoDs sans pre-amplification. The great deal of confusion around reporting of CRISPR-Cas kinetic rates and LoDs is currently of such magnitude and prevalence that it limits our ability to assess these enzyme systems as a practical candidate for diagnostic assays. One thing is certain: The development of the CRISPR-Cas diagnostics field so far suggests self-reflection by the community and urges all of us to proceed with caution.

**Acknowledgements.** The author gratefully acknowledges the valuable advice provided by Professor Richard Zare of the Chemistry Department of Stanford University on a draft of this paper. The author also gratefully acknowledges support from the Stanford Bio-X Interdisciplinary Initiatives Committee (IIP) [R10-application 55] [Principal investigators J.G.S. and Niaz Banaei of Stanford University].

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
