## [Reviewer Report]

*Comments to Author*: This Perspective analyzes the current state of the CRISPR diagnostic field that is growing rapidly and is finding application as a sensitive diagnostic tool to detect a wide range of diseases.However, there are a number of inconsistences in the literature. The author states “It is the intension of this Perspective article to identify potential improvements in the manner in which CRISPR kinetic rates and assay LoDs are reported and compared.”

The author has carefully scanned literature and found many inconsistences.He directly wrote to the key lab that published the papers and alerted them these inconsistences. As a result, these labs made corrections and published errata and acknowledged this author careful observations.The author is now sounding alarm to other labs in general to pay attentions to the inconsistences since such inconsistences not only are wasteful to others’ efforts, but also generate irreproducible results, generating confusion in the literature.It is important to point out and alert the people who carry out such analysis using CRISPR methods.

Some minor points:

1) Since there are several pros and cons in the CRISPR methods, it is best to summarize in a Table to list each Pro and Con and compare them directly.The inconsistence in the published literature should also summarized in the Table, or in a separate table.

2) If necessary, this reviewer suggests to add another figure to illustrate the possible errors and key steps that result these inconsistences and how to avoid them in the future. A picture is worth 10,000 words.

After the minor revision, this reviewer recommends publication as a Perspective in QRB Discovery.

---

## [Reviewer Report]

*Comments to Author*: The reviewer strongly recommends accepting the manuscript “Inconsistent Treatments of CRISPR Kinetics Impair Assessment of Its Diagnostic Potential” for publication (with minor revisions) as a Perspective. The author Juan G. Santiago shows how the enzyme turnover rates of well-known CRISPR enzymes have erroneously been overstated by several orders of magnitude by a few seminal papers. Although errata were published for some of them, several later authors/papers are now corroborating each other on reporting unrealistic turnover rates. The manuscript by Santiago therefore proposes several rules of thumb as a self-consistency check of enzyme kinetic data, based on simple conservation of mass and Michaelis Menten kinetics, which should prove quite useful for researchers and authors.

Considering the high and increasing popularity of diagnostics/biosensors based on CRISPR-Cas systems, every effort should be made to report correct CRISPR-Cas reaction rates. Santiago shows several examples of existing/prototype sensors which are probably based on questionable kinetics. The urgency of the manuscript can therefore be rated as high.

A few minor points:

1. Especially in the abstract and introduction, it should be clarified that Cas13 cuts RNA and not DNA.

2. The three formulae (especially alpha, beta) are given as ratios which are to be less than 1. This is a matter of personal taste, but absent any special reason, it would perhaps be clearer to the reader to omit alpha and beta, and instead rewrite as numerator < denominator. Alternatively, maybe some additional explaining text could help.

3. The process of deriving the third formula (gamma) should perhaps be presented (like, substituting rate = kcat/KM * E0 * S). Otherwise, the reader lacks an adequate explanation.

4. The erratum for Slaymaker (2019) appears to be published in 2019, before the main article in 2021. The reference for Ramachandran 2021c also seems incomplete.

---

## [Reviewer Report]

*Comments to Author*: Reviewer #1: The reviewer strongly recommends accepting the manuscript “Inconsistent Treatments of CRISPR Kinetics Impair Assessment of Its Diagnostic Potential” for publication (with minor revisions) as a Perspective. The author Juan G. Santiago shows how the enzyme turnover rates of well-known CRISPR enzymes have erroneously been overstated by several orders of magnitude by a few seminal papers. Although errata were published for some of them, several later authors/papers are now corroborating each other on reporting unrealistic turnover rates. The manuscript by Santiago therefore proposes several rules of thumb as a self-consistency check of enzyme kinetic data, based on simple conservation of mass and Michaelis Menten kinetics, which should prove quite useful for researchers and authors.

Considering the high and increasing popularity of diagnostics/biosensors based on CRISPR-Cas systems, every effort should be made to report correct CRISPR-Cas reaction rates. Santiago shows several examples of existing/prototype sensors which are probably based on questionable kinetics. The urgency of the manuscript can therefore be rated as high.

A few minor points:

1. Especially in the abstract and introduction, it should be clarified that Cas13 cuts RNA and not DNA.

2. The three formulae (especially alpha, beta) are given as ratios which are to be less than 1. This is a matter of personal taste, but absent any special reason, it would perhaps be clearer to the reader to omit alpha and beta, and instead rewrite as numerator < denominator. Alternatively, maybe some additional explaining text could help.

3. The process of deriving the third formula (gamma) should perhaps be presented (like, substituting rate = kcat/KM * E0 * S). Otherwise, the reader lacks an adequate explanation.

4. The erratum for Slaymaker (2019) appears to be published in 2019, before the main article in 2021. The reference for Ramachandran 2021c also seems incomplete.

Reviewer #2: This Perspective analyzes the current state of the CRISPR diagnostic field that is growing rapidly and is finding application as a sensitive diagnostic tool to detect a wide range of diseases.However, there are a number of inconsistences in the literature. The author states “It is the intension of this Perspective article to identify potential improvements in the manner in which CRISPR kinetic rates and assay LoDs are reported and compared.”

The author has carefully scanned literature and found many inconsistences.He directly wrote to the key lab that published the papers and alerted them these inconsistences. As a result, these labs made corrections and published errata and acknowledged this author careful observations.The author is now sounding alarm to other labs in general to pay attentions to the inconsistences since such inconsistences not only are wasteful to others’ efforts, but also generate irreproducible results, generating confusion in the literature.It is important to point out and alert the people who carry out such analysis using CRISPR methods.

Some minor points:

1) Since there are several pros and cons in the CRISPR methods, it is best to summarize in a Table to list each Pro and Con and compare them directly.The inconsistence in the published literature should also summarized in the Table, or in a separate table.

2) If necessary, this reviewer suggests to add another figure to illustrate the possible errors and key steps that result these inconsistences and how to avoid them in the future. A picture is worth 10,000 words.

After the minor revision, this reviewer recommends publication as a Perspective in QRB Discovery.